# Curated single cell multimodal landmark datasets for R/Bioconductor

**Kelly B. Eckenrode**[1,2‡], **Dario Righelli**[3‡], **Marcel Ramos**[1,2,4‡], **Ricard Argelaguet**[5‡], **Christophe Vanderaa**[6‡], **Ludwig Geistlinger**[7‡], **Aedin C. Culhane**[8], **Laurent Gatto**[6], **Vincent Carey**[9], **Martin Morgan**[4], **Davide Risso**[3]*, **Levi Waldron**[1,2]*

**1** Graduate School of Public Health and Health Policy, City University of New York, NY, NY, United States of America, **2** Institute for Implementation Science in Public Health, City University of New York, NY, NY, United States of America, **3** Department of Statistical Sciences, University of Padova, Padova, Italy, **4** Roswell Park Comprehensive Cancer Center, Buffalo, New York, United States of America, **5** European Bioinformatics Institute (EMBL-EBI), Hinxton, Cambridgeshire, United Kingdom, **6** de Duve Institute, Université catholique de Louvain, Brussels, Belgium, **7** Center for Computational Biomedicine, Harvard Medical School, Boston, Massachusetts, United States of America, **8** School of Medicine, University of Limerick, Limerick, Ireland, **9** Channing Division of Network Medicine, Brigham and Women's Hospital and Harvard Medical School, Boston, Massachusetts, United States of America

‡ KBE, DR, MR share first authorship on this work. RA, CV, LG share second authorship on this work.
* davide.risso@unipd.it (DR); levi.waldron@sph.cuny.edu (LW)

**Data Availability Statement:** The data reviewed and curated in this review are publicly available under the Artistic 2.0 license as the SingleCellMultiModal Bioconductor package

## Abstract

### Background

The majority of high-throughput single-cell molecular profiling methods quantify RNA expression; however, recent multimodal profiling methods add simultaneous measurement of genomic, proteomic, epigenetic, and/or spatial information on the same cells. The development of new statistical and computational methods in Bioconductor for such data will be facilitated by easy availability of landmark datasets using standard data classes.

### Results

We collected, processed, and packaged publicly available landmark datasets from important single-cell multimodal protocols, including CITE-Seq, ECCITE-Seq, SCoPE2, scNMT, 10X Multiome, seqFISH, and G&T. We integrate data modalities via the *MultiAssayExperiment* Bioconductor class, document and re-distribute datasets as the *SingleCellMultiModal* package in Bioconductor's Cloud-based *ExperimentHub*. The result is single-command actualization of landmark datasets from seven single-cell multimodal data generation technologies, without need for further data processing or wrangling in order to analyze and develop methods within Bioconductor's ecosystem of hundreds of packages for single-cell and multimodal data.

### Conclusions

We provide two examples of integrative analyses that are greatly simplified by *SingleCellMultiModal*. The package will facilitate development of bioinformatic and statistical methods in Bioconductor to meet the challenges of integrating molecular layers and analyzing phenotypic outputs including cell differentiation, activity, and disease.

(https://doi.org/doi:10.18129/B9.bioc.
SingleCellMultiModal), with open development and
issue tracking on Github (https://github.com/
waldronlab/SingleCellMultiModal). The original 10X
Genomics Multiome data are available from https://
support.10xgenomics.com/single-cell-multiome-
atac-gex/datasets.

**Funding:** This research was supported in part by
the National Cancer Institute of the National
Institutes of Health (2U24CA180996) to DRis,
DRig, VC, MM, MR, KE, LG, and LW, and by the
Chan Zuckerberg Initiative DAF (CZF2019-002443),
an advised fund of Silicon Valley Community
Foundation to DRis, DRig, MM. CV was supported
by a PhD fellowship from the Belgian National Fund
for Scientific Research (FNRS). The funders had no
role in study design, data collection and analysis,
decision to publish, or preparation of the
manuscript.

**Competing interests:** The authors have declared
that no competing interests exist.

## Author summary

Experimental data packages that provide landmark datasets have historically played an
important role in the development of new statistical methods in Bioconductor by lowering
the barrier of access to relevant data, providing a common testing ground for software
development and benchmarking, and encouraging interoperability around common data
structures. In this manuscript, we review major classes of technologies for collecting mul-
timodal data including genomics, transcriptomics, epigenetics, proteomics, and spatial
information at the level of single cells. We present the SingleCellMultiModal R/Biocon-
ductor package that provides single-command access to landmark datasets from seven dif-
ferent technologies, storing datasets using HDF5 and sparse arrays for memory efficiency
and integrating data modalities via the MultiAssayExperiment class. We demonstrate two
integrative analyses that are greatly simplified by SingleCellMultiModal. The package
facilitates development and benchmarking of bioinformatic and statistical methods to
integrate molecular layers at the level of single cells with phenotypic outputs including cell
differentiation, activity, and disease, within Bioconductor's ecosystem of hundreds of
packages for single-cell and multimodal data.

This is a *PLOS Computational Biology* Benchmarking paper.

## Introduction

Understanding the quantitative relationship between molecules and physiology has motivated
the development of quantitative profiling techniques, especially for single-cell sequencing [1].
Single-cell multimodal omics technologies (Nature Method of the Year 2019 [2]) couple sin-
gle-cell RNA sequencing with other molecular profiles such as DNA sequences, methylation,
chromatin accessibility, cell surface proteins, and spatial information, simultaneously in the
same cell. Integrative analysis of multiple molecular measurements from the same cell has
enabled, for example, discovery of rare cell types by defining subpopulations based on surface
markers with CITE-Seq [3] and ECCITE-Seq [4] (Cellular Indexing of Transcriptomes and
Epitopes by sequencing, Expanded CRISPR CITE-Seq), of epigenetic regulation and cell differ-
entiation lineage with scNMT-seq [5] (single-cell nucleosome, methylation, and transcriptome
sequencing), a high resolution commercial version of single cell chromatin accessibility with
10X Multiomics [6], understanding of spatial patterns of gene expression with seq-FISH [7],
and correlation of genotype-phenotype in healthy and disease states with G&T-seq [8] (parallel
Genome and Transcriptome sequencing). Other single-cell multimodal datasets take measure-
ments from separate cells due to the technical constraints, like mass-spectrometry based prote-
omic methods including SCoPE2 [9] (single-cell protein analysis by mass spectrometry).

Capturing and integrating an array of different molecular signals at the single-cell level
poses new analytical challenges. Single-cell multimodal experiments generate multidimen-
sional and high volume datasets, requiring distinct informatic and statistical methods to store,
process and analyze data. Integrating different molecular layers to provide biologically mean-
ingful insight is an active area of development in R/Bioconductor due to the availability of data
containers and analysis toolkits for single-cell analysis. R/Bioconductor is an open develop-
ment and open source platform for analyzing biomedical and genomic data with dedicated

data structures such as the *SingleCellExperiment* class [10] for single-cell data and the *MultiAssayExperiment* class [11] for multi-omics data. Both are designed based on the *SummarizedExperiment* class [12], the central Bioconductor data structure for storing, manipulating, and analyzing high-throughput quantitative omics data. Relative to analysis platforms within and outside of the R programming language (e.g. GATK, Seurat [13], mixOmics [14], MOFA+ [15], CiteFuse package [16], ScanPy for CITE-Seq [17], Conos for SCoPE2 [18]), Bioconductor provides the broadest range of interoperable data structures and packages for statistical analysis and visualization of single-cell multimodal data.

Easy availability of publicly available experimental data using standardized data classes has long played an important role in the development of interoperable software packages for the analysis of data from new technologies, helping to coalesce development efforts around shared datasets and commonly used data classes such as *ExpressionSet* [19] and then *(Ranged)SummarizedExperiment* [20] and *SingleCellExperiment* [10]. We therefore introduce a suite of single-cell multimodal landmark datasets for benchmarking and testing multimodal analysis methods via the Bioconductor ExperimentHub package *SingleCellMultiModal* (Fig 1A). The scope of this package is to provide efficient access to a selection of curated, pre-integrated, publicly available landmark datasets for methods development and benchmarking within the Bioconductor ecosystem. Also, we included cell type labels using ontology terms for each experimental database called ontomap (see Methods). Cell labels metadata helps users develop a common ground truth. Some such methods and code for analysis workflows are reviewed by Lê Cao et al. [21]. Users can obtain integrative representations of multiple modalities as a *MultiAssayExperiment*, a common core Bioconductor data structure relied on by dozens of multimodal data analysis packages. Each dataset was quality controlled; either by the original authors during publication, or we implemented a quality control pre-filtered for high quality cells. *SingleCellMultiModal* uses Bioconductor's *ExperimentHub* package and service to host, coordinate, and manage the data from the cloud. We plan to update the package as new datasets and technologies become available and we welcome community contributions. This manuscript serves as a review of essential aspects of these technologies suitable for developers of bioinformatic and statistical software, and as a description of the *SingleCellMultiModal* data package.

## Results

### Summary of landmark datasets in *SingleCellMultiModal*

To evaluate and design new statistical methods that accompany experimental single-cell multimodal data, it is important to establish landmark datasets. The goal of this section is to provide an overview of the landmark datasets currently in *SingleCellMultiModal* as well as to introduce the experimental and technological context for each experimental assay (Table 1). For more information concerning the details of the technologies, consult [22]. We briefly describe each landmark experiment including context, major findings from the publication, and challenges in its analysis, then summarize its accompanying dataset in *SingleCellMultiModal* including number of cells and features (Fig 1B).

**RNA and protein: antibody tagged cell surface markers.   Purpose and goals:** Traditionally, protein expression in cell populations are measured using flow cytometry. With the advent of single-cell multimodal methods cell surface proteins are measured with higher resolution with simultaneous measurements of mRNA abundance, which enhances the ability to identify new cell subpopulations in heterogeneous samples. Cellular Indexing of Transcriptomes and Epitopes by sequencing (CITE-Seq) measures protein cell surface markers and gene expression in the same cell. An extension of CITE-Seq is ECCITE-Seq, Expanded CRISPR-

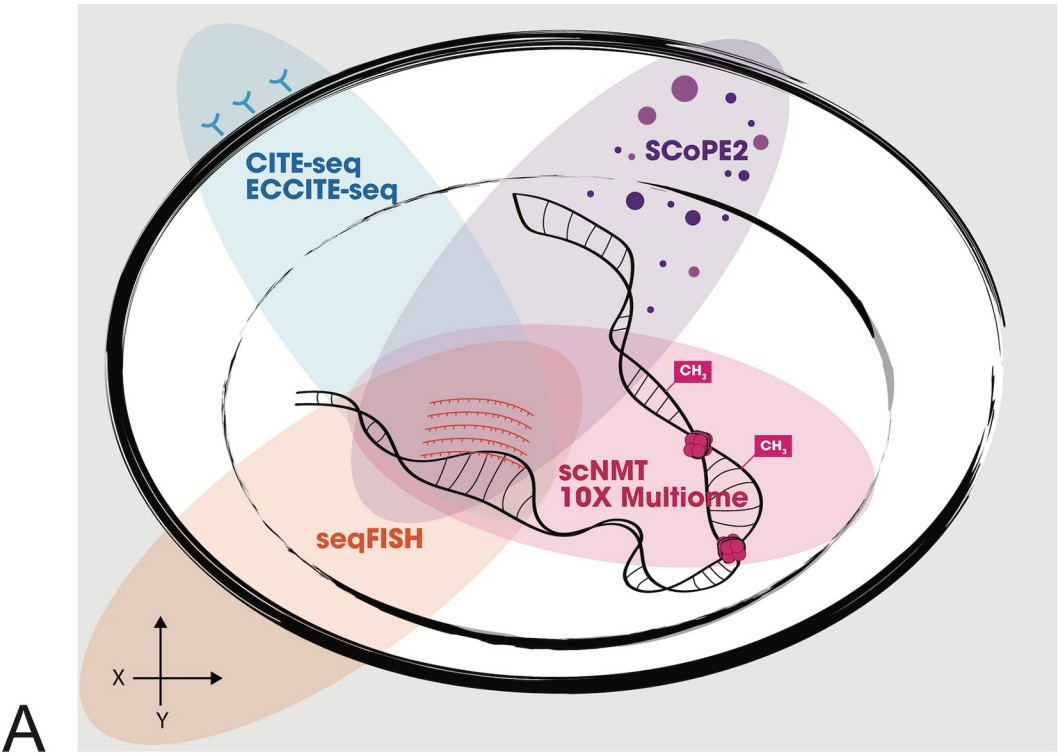

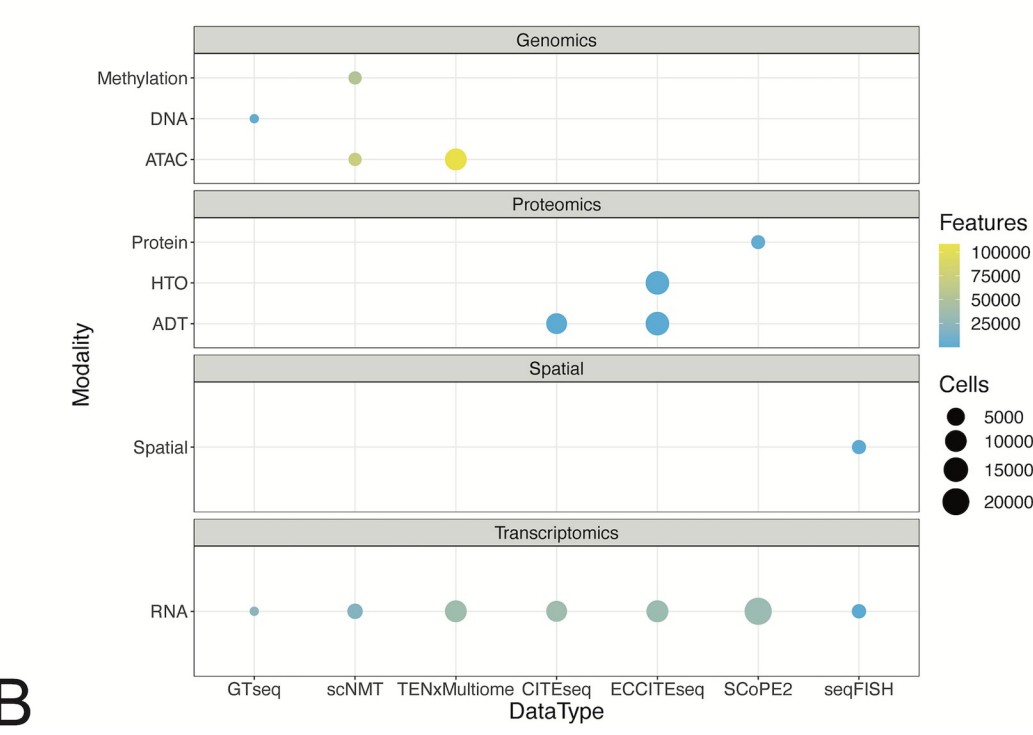

**Fig 1. Representation of modalities included in the SingleCellMultiModal package.** (A) a Venn diagram representation of the modalities collected by each different technology, including: RNA (center), surface proteins (top left), spatial information (bottom left), methylation and open chromatin (bottom right), and peptides (top right). (B) The number of features and cells collected for each data modality by each technology.

**Table 1. Single-cell multimodal datasets included *SingleCellMultiModal* package.** Modalities refer to the molecular feature measured in the experimental assay. Cell/ process type provides information on the type of material or development event data was collected. Datatype name column refers to the dataset name in *SingleCellMultiModal*.

| MODALITIES | EXPERIMENTAL ASSAY | CELL / PROCESS TYPE | DATATYPE NAME | CITATION |
|---|---|---|---|---|
| RNA + DNA | G&T-seq | Mouse epithelial, human breast tumor | mouse_embryo_8_cell | [8] |
| RNA + Protein | CITE-Seq | Cord blood mononuclear | cord_blood | [3] |
| | ECCITE-Seq | Peripheral blood mononuclear, human T-cell lymphoma, mouse fibroblast | peripheral_blood | [4] |
| RNA + Epigenetic | scNMT-seq | Mouse gastrulation | mouse_gastrulation | [23] |
| | 10X Multiome scATAC-seq + Single-cell RNA-seq | Peripheral blood mononuclear | pbmc_10x | [6] |
| RNA + Spatial | seqFISH | Mouse cortical neuronal | mouse_visual_cortex | [24],[22] |
| RNA + Proteomic | SCoPE2 | Human monocyte and PMA-induced macrophage | macrophage_differentiation | [9] |

compatible CITE-Seq, which allows for the capture of sgRNA from CRISPR mediated screens. Collectively, these technologies provide a high-throughput method for single-cell immunophenotyping and transcriptome analysis.

**Technology:** CITE-Seq relies on antibodies conjugated to DNA barcodes to infer protein levels, and in tandem count DNA handles from PCR amplification of mRNA transcripts. Inside the droplet contains mRNA transcripts, proteins conjugated with antibody derived tags (ADTs), beads decorated with oligo-dT, reverse transcriptase and primers for cDNA amplification. The use of DNA barcodes is a departure from traditional fluorescence labels, which are limited in number because of the overlaps in spectral detection, excitation and emission frequencies [25].

A variation of CITE-Seq is ECCITE-Seq that can track single-cell CRISPR screens using sgRNA sequencing capture [4]. The CRISPR-Cas9 system is used to generate targeted gene knockout/mutants by using two components: sgRNA (single guide RNA for gene of interest) and Cas9 (endonuclease for cleaving double DNA strand breaks). sgRNA are composed of custom crRNA 17-22nt with a scaffold tracrRNA, which means the sgRNA are composed of two RNA pieces: one is customizable and the other is not. The sgRNA targets the gene of interest and orchestrates the Cas9 enzyme to gene location to insert a variety of mutations or full gene knock-outs. The CRISPR-Cas9 system introduces targeted gene mutations with greater ease at the bench, plus it is easier to scale up to many more experimental samples than previous approaches.

Antibody oligo counts are listed in the ADT and HTO (hashtag oligo) tables and sgRNAs counts in the GDO tables. After cell perturbations via CRISPR screens, cells are collected and prepared with 10X Genomics V(D)J solution which incorporates Single-cell RNA-seq with additional profiling of protein surface markers and sgRNAs (when applicable). The molecular contents, mRNA and DNA-tagged proteins, will hybridize to the decorated beads. The benefit of adding barcoding to cells is that it allows for tracking of doublets (two cells in one droplet).

**Landmark data:** There are several experimental datasets derived from the original CITE-Seq landmark paper. Among them we selected the cord blood dataset where the cells have been incubated with CITE-seq antibody conjugates and fluorophore-conjugated antibodies. This *cord_blood* dataset has two different assays. The scADT assay is a matrix indicating the 13 proteins surface abundance for each of the 8617 cells, while the scRNA assay is a matrix of 20400 human genes and 15880 mouse genes where each entry contains the expression abundance in each of the 8617 cells (Table 2).

The package also includes an ECCITE-Seq dataset aimed at characterizing immune subpopulation cell types after an experimental perturbation. The *peripheral_blood* dataset is

**Table 2. CITE-Seq dataset description, with assay types, molecular modes, number of specimens, number of features and number of cells.** ADTs, antibody derived tags.

| Dataset Identifier | Assay Type | Modes | Species | Data Structure | Version | # features | # cells |
|---|---|---|---|---|---|---|---|
| Cord blood | RNA-seq | Transcripts | Human | matrix | 1.0.0 | 36280 | 8617 |
| | ADT | Proteins | Human | matrix | 1.0.0 | 13 | 8617 |

organized in two different conditions: the control (CTRL) and the cutaneous T-cell lymphoma (CTCL). For both conditions the ECCITE-Seq protocol has been performed to produce transcripts (RNA-seq), proteins (ADT) and cell tracking (HTO) abundance. All these modalities are collected as separated assays into the *MultiAssayExperiment*, where a sparse matrix is used to store the RNA-seq counts. The modalities are collected from the same cells, but not all the cells are entirely profiled by the same modalities. Of the total 36248 cells, 4190 cells from the CTCL and 4292 cells from the CTRL are matched with all modalities (Fig 2). sgRNA data is stored in long format providing access through the metadata data structure of the *MultiAssayExperiment*. The CITE-Seq dataset is accessible via the *SingleCellMultiModal* package by using the `CITEseq(DataType="cord_blood")` function call, while for the ECCITE-Seq data it's sufficient to change the identifier as follow `CITEseq(DataType="peripheral_blood")`. Both function calls return a *MultiAssayExperiment* object with matrices or sparse matrices as assays (Table 3).

**RNA and protein: mass spectrometry-based. Purpose and goals:** CITE-Seq offers valuable information about the expression of surface proteins. However, the acquisition is limited to tens of targets as the identification relies on antibodies. Furthermore, it cannot provide information on intracellular markers. Mass spectrometry (MS)-based single-cell proteomics (SCP) provides a means to overcome these limitations and to perform unbiased single-cell profiling of the soluble proteome. MS-SCP is emerging thanks to recent advances in sample preparation, liquid chromatography (LC) and MS acquisition. The technology is in its infancy and protocols still need to be adapted in order to acquire multiple multimodalities from a single-cell. In this section the multimodality is achieved by subjecting similar samples to MS-SCP and Single-cell RNA-seq.

**Technology:** The current state-of-the-art protocol for performing MS-SCP is the SCoPE2 protocol [9]. Briefly, single-cells are lysed, proteins are extracted and digested into peptides. The peptides are then labeled using tandem mass tags (TMT) in order to multiplex up to 16 samples per run (Fig 3A). The pooled peptides are then analysed by liquid chromatography-tandem mass spectrometry (LC-MS/MS). LC separates the peptides based on their mass and affinity for the chromatographic column. The peptides are immediately ionized as they come out (Fig 3B) and are sent for two rounds of MS (MS/MS, Fig 3C). The first round isolates the ions based on their mass to charge (m/z) value. The isolated ions are fragmented and sent to the second round of MS that records the m/z and intensity of each fragment. The pattern of intensities over m/z value generated by an ion is called an MS2 spectrum. The MS2 spectra are then computationally matched to a database to identify the original peptide sequence from which they originated. The spectra that were successfully associated to a peptide sequence are called peptide to spectrum matches (PSMs, Fig 3D). Next to that, a specific range of the MS spectrum holds the TMT label information where each label generates a fragment with an expected m/z value. The intensity of each label peak is proportional to the peptide expression in the corresponding single cell and this allows for peptide quantification (Fig 3D). Finally, the quantified PSM data go through a data processing pipeline that aims to reconstruct the protein data that can be used for downstream analyses (Fig 3E).

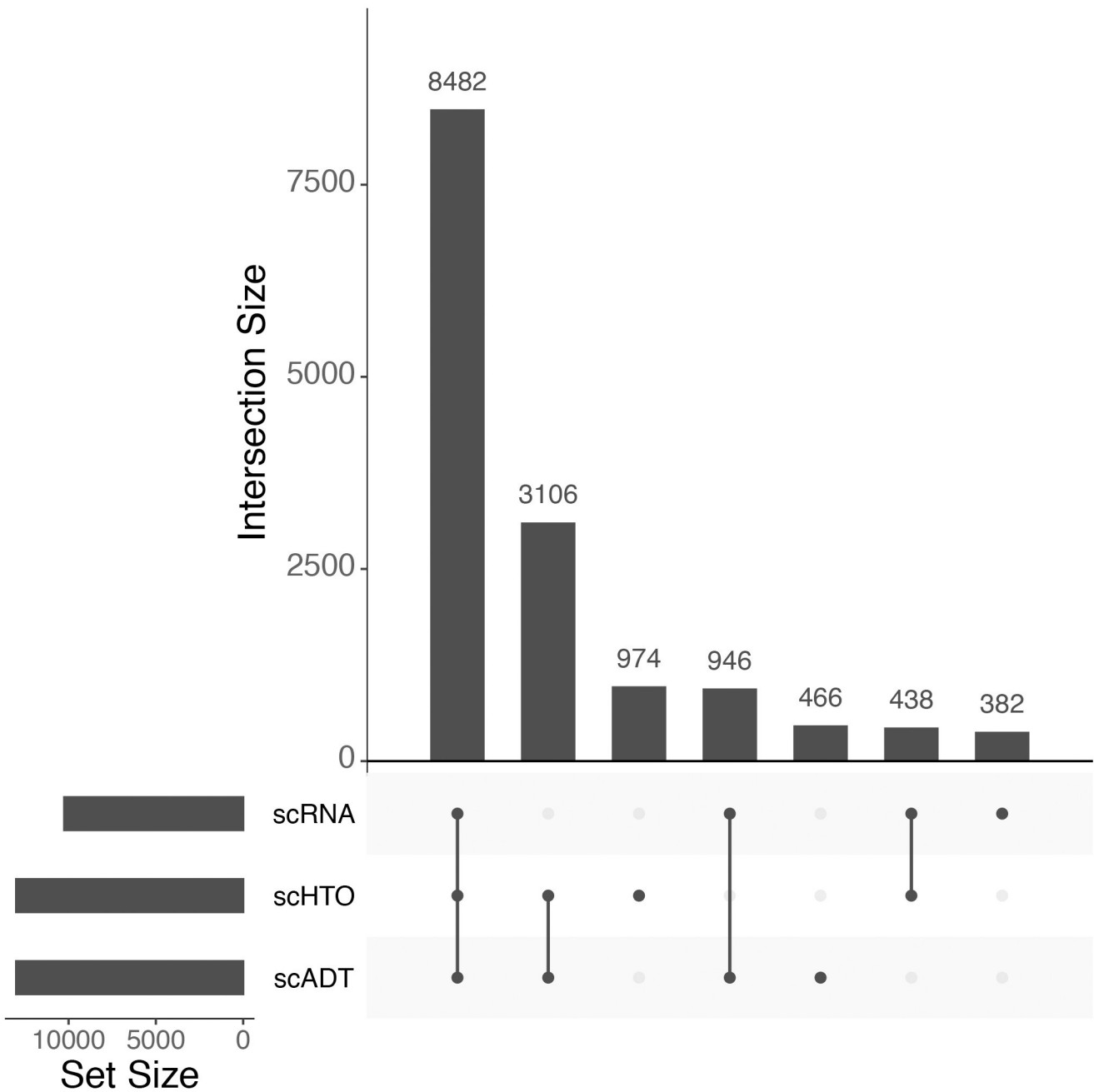

**Fig 2. Upset plot [26] of the overlap of modalities on the same cells in the control sample of the ECCITE-Seq "peripheral blood" dataset.** 8482 cells are assayed in all three modes (ADT, HTO, RNA), 3105 cells are assayed by HTO and ADT only, etc. RNA data are available for ~10248 cells, whereas HTO and ADT data are each individually available for ~13000 cells across both conditions. This plot is produced by the upsetSamples function of the *MultiAssayExperiment* package, and can be applied directly to all datasets produced by *SingleCellMultiModal*.

The major challenge in MS-SCP is to recover sufficient peptide material for accurate peptide identification and quantification. SCoPE2 solves this issue by optimizing the sample preparation step to limit samples loss, by providing analytical tools to optimize the MS/MS settings, and most importantly by introducing a carrier sample into the pool of multiplexed samples. The carrier is a sample that contains hundreds of cells instead of a single-cell and allows to

**Table 3. ECCITE-Seq dataset description: assay types, molecular modes, number of specimens, number of features and number of cells.** ADTs, antibody derived tags; HTO, Hashtagged oligos; sgRNAs, CRISPR V(D)J's.

| Dataset Identifier | Assay Type | Modes | Species | Data Structure | Condition | # features | # cells |
|---|---|---|---|---|---|---|---|
| Peripheral blood | RNA-seq | Transcripts | Human | dgCMatrix | CTCL | 33538 | 5399 |
| | | | | | CTRL | 33538 | 4849 |
| | ADT | Proteins | Human | dgCMatrix | CTCL | 52 | 6500 |
| | | | | | CTRL | 52 | 6500 |
| | HTO | Cell tracking | Human | dgCMatrix | CTCL | 7 | 6500 |
| | | | | | CTRL | 7 | 6500 |
| | sgRNAs stored in long format | | | | | # rows | # cols |
| | sgRNAs | CRISPR perturbation | Human | data.frame | CTCL TCRab | 9626 | 18 |
| | | | | | CTCL TCRgd | 2430 | 18 |
| | | | | | CTRL TCRab | 8359 | 18 |
| | | | | | CTRL TCRgd | 3099 | 18 |

boost the peptide identification rate by increasing the amount of peptide material delivered to the MS instrument.

Parallel to SCoPE2, other groups have developed a label-free MS-SCP, where each LC-MS/MS run contains unlabelled peptides from a single cell [27]. Although it allows for more accurate quantifications, it suffers from low throughput. The current methodological advances in MS-SCP have extensively been reviewed elsewhere [28].

**Landmark data:** The SCoPE2 dataset we provide in this work was retrieved from the supplementary information of the landmark paper [9]. This is a milestone dataset as it is the first publication where over a thousand cells are measured by MS-SCP. The research question is to understand whether a homogeneous monocyte population (U-937 cell line) could differentiate upon PMA treatment into a heterogeneous macrophage population, namely whether M1 and M2 macrophage profiles could be retrieved in the absence of differentiation cytokines. Different replicates of monocyte and macrophage samples were prepared and analyzed using either MS-SCP or Single-cell RNA-seq. The MS-SCP data was acquired in 177 batches with on average 9 single-cells per batch. The Single-cell RNA-seq data was acquired in 2 replicates with on average 10,000 single-cells per acquisition using the 10x Genomics Chromium platform. Cell type annotations are only available for the MS-SCP data. Note also that MS-SCP data provides expression information at protein level meaning that the peptide data has already been processed. The processing includes filtering high quality features, filtering high quality cells, log-transformation, normalization, aggregation from peptides to proteins, imputation and batch correction (Fig 3E). More details on the protein data processing can be found in the original paper or in the paper that reproduced that analysis [29]. Count tables were provided for the Single-cell RNA-seq dataset with no additional processing.

The data can be accessed in the *SingleCellMultiModal* package by calling `SCoPE2` (`"macrophage_differentiation"`) (Table 4). Relevant cell metadata is provided within the MultiAssayExperiment object. The MS-SCP dataset contains expression values for 3,042 proteins in 1,490 cells. The Single-cell RNA-seq contains expression values for 32,738 genes (out of which 10,149 are zero) for 20,274 cells.

**Single-cell nucleosome, methylation and transcription sequencing (scNMT-seq). Purpose and goals:** The profiling of the epigenome at single-cell resolution has received increasing interest, as it provides valuable insights into the regulatory landscape of the genome [30,31]. Although the term epigenome comprises multiple molecular layers, the profiling of chromatin accessibility and DNA methylation have received the most attention to date.

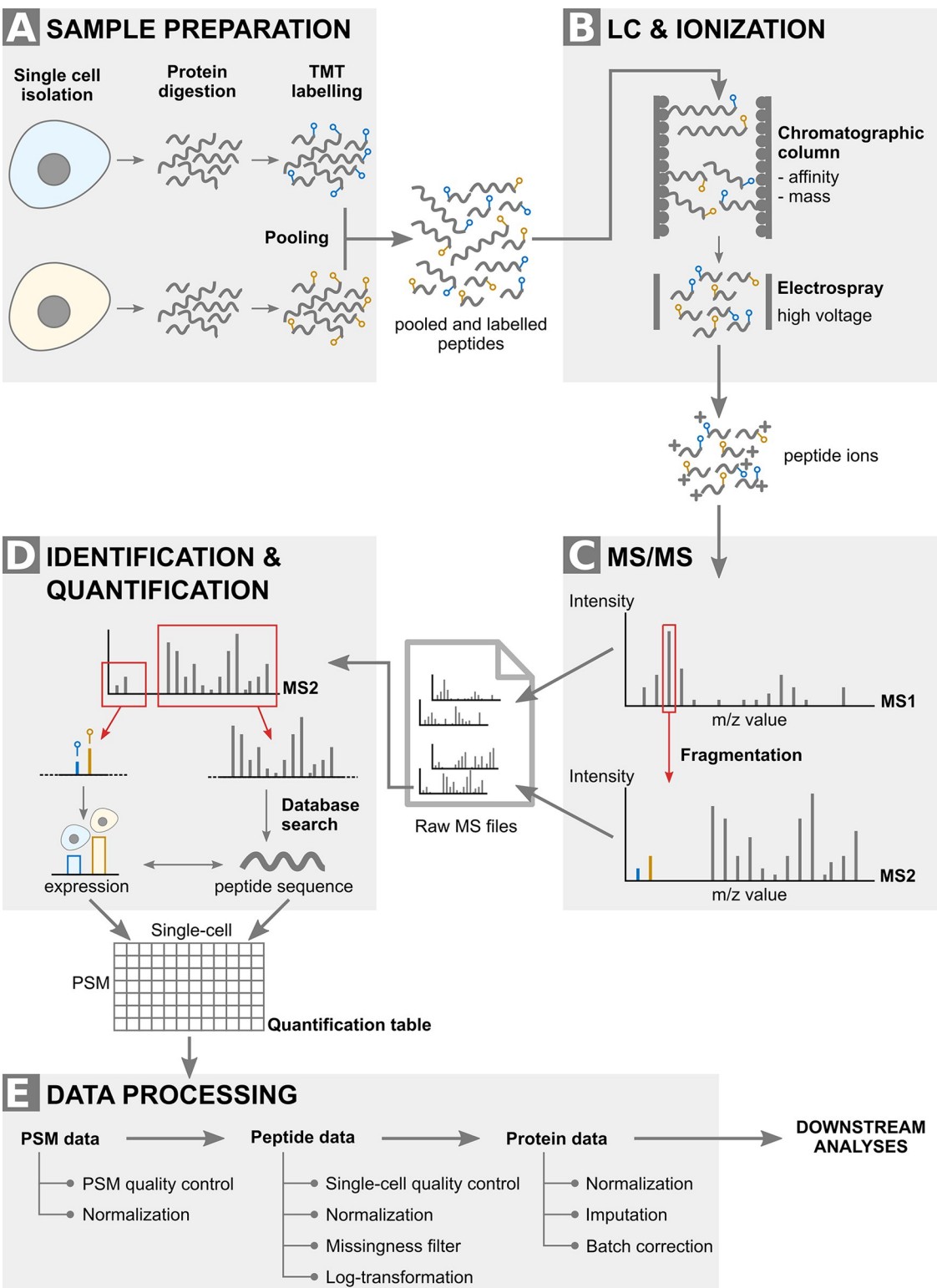

**Fig 3. SCoPE2 workflow.** The workflow consists of 4 main steps. (A) Sample preparation extracts and labels peptides from single-cells. (B) LC separates the peptides based on their mass and affinity for the column. Note that the TMT tag does not influence those properties. Peptides that are eluting are ionised thanks to an electrospray. (C) MS/MS performs an m/z scan of the incoming ions to select the most abundant ones that are then fragmented separately. A second round of MS acquires the spectrum generated by the ion fragments. (D) Each spectrum is then computationally processed to obtain the cell-specific expression values and the peptide

identity. (E) The data processing pipeline reconstructs the protein data from the quantified PSMs. Abbreviations: TMT: tandem mass tags; LC: liquid chromatography; MS: mass spectrometry; MS/MS: tandem MS; m/z: mass over charge; PSM: peptide to spectrum match.

**Technology:** DNA methylation is generally measured using single-cell bisulfite sequencing (scBS-seq) [32]. The underlying principle of scBS-seq is the treatment of the DNA with sodium bisulfite before DNA sequencing, which converts unmethylated cytosine (C) residues to uracil (and after retro-PCR amplification, to thymine (T)), leaving 5-methylcytosine residues intact. The resulting C→T transitions can then be detected by DNA sequencing. Further methodological innovations enabled DNA methylation and RNA expression to be profiled from the same cell, demonstrated by the scM&T-seq assay [33].

Chromatin accessibility was traditionally profiled in bulk samples using DNase sequencing (DNase-seq) [34]. However, in recent years, transposase-accessible chromatin followed by sequencing (ATAC-seq) has displaced DNase-seq as the *de facto* method for profiling chromatin accessibility due to its fast and sensitive protocol, most notably in single-cell genomics [35]. Briefly, in ATAC-seq, cells are incubated with a hyperactive mutant Tn5 transposase, an enzyme that inserts artificial sequencing adapters into nucleosome-free regions. Subsequently, the adaptors are purified, PCR-amplified and sequenced. Notably, single-cell ATAC-seq has also been combined with Single-cell RNA-seq to simultaneously survey RNA expression and chromatin accessibility from the same cell, as demonstrated by SNARE-seq [36], SHARE-seq [37] and the recently commercialized Multiome Kit from 10x Genomics [6]. Finally, some assays have been devised to capture at least three molecular layers from the same cell, albeit at a lower throughput than SNARE-seq or SHARE-seq. An example is scNMT-seq (single-cell nucleosome methylation and transcriptome sequencing) [5]. scNMT captures a snapshot of RNA expression, DNA methylation and chromatin accessibility in single-cells by combining two previous multi-modal protocols: scM&T-seq [33] and Nucleosome Occupancy and Methylation sequencing (NOMe-seq) [38]

In the first step (the NOMe-seq step), cells are sorted into individual wells and incubated with a GpC methyltransferase. This enzyme labels accessible (or nucleosome depleted) GpC sites via DNA methylation. In mammalian genomes, cytosine residues in GpC dinucleotides are methylated at a very low rate. Hence, after the GpC methyltransferase treatment, GpC methylation marks can be interpreted as direct readouts for chromatin accessibility, as opposed to the CpG methylation readouts, which can be interpreted as endogenous DNA methylation. In a second step (the scM&T-seq step), the DNA molecules are separated from the mRNA using oligo-dT probes pre-annealed to magnetic beads. Subsequently, the DNA fraction undergoes scBS, whereas the RNA fraction undergoes Single-cell RNA-seq.

**Landmark data:** The scNMT landmark paper reported simultaneous measurements of chromatin accessibility, DNA methylation, and RNA expression at single-cell resolution during early embryonic development, spanning exit from pluripotency to primary germ layer specification [23]. This dataset represents the first multi-omics roadmap of mouse gastrulation at single-cell resolution. Using multi-omic integration methods, the authors detected genomic associations between distal regulatory regions and transcription activity, revealing novel insights into the role of the epigenome in regulating this key developmental process.

**Table 4. SCoPE2 dataset descriptions, with assay types, molecular modes, specimens, dataset version provided, number of features and number of cells.**

| Dataset Identifier | Assay Type | Modes | Species | Data Structure | Version | # features | # cells |
|---|---|---|---|---|---|---|---|
| macrophage _differentiation | LC-MS/MS | Proteins | Human | matrix | 1.0.0 | 3,042 | 1,490 |
| | RNA-seq | Transcripts | Human | HDF5 | 1.0.0 | 32,738 | 20,274 |

One of the challenges of this dataset is the complex missing value structure. Whereas RNA expression is profiled for most cells (N = 2480), DNA methylation and chromatin accessibility is only profiled for subsets of cells (N = 986 and N = 1105, respectively). This poses important challenges to some of the conventional statistical methods that do not handle missing information.

The output of the epigenetic layers from scNMT-seq is a binary methylation state for each observed CpG (endogenous DNA methylation) and GpC (a proxy for chromatin accessibility). However, instead of working at the single nucleotide level, epigenetic measurements are typically quantified over genomic features (i.e. promoters, enhancers, etc.). This is done assuming a binomial model for each cell and feature, where the number of successes is the number of methylated CpGs (or GpCs) and the number of trials is the total number of CpGs (or GpCs) that are observed. Here we provide DNA methylation and chromatin accessibility estimates quantified over CpG islands, gene promoters, gene bodies and DNAse hypersensitive sites (defined in Embryonic Stem Cells).

The pre-integrated scNMT dataset is accessed from the *SingleCellMultiModal* package by calling e.g. scNMT("mouse_gastrulation", version = "1.0.0") (Table 5). Relevant cell metadata is provided within the MultiAssayExperiment object. The overall dataset is 277MB.

**Chromium Single-cell Multiome ATAC and gene expression.   Purpose and goals:** A new commercial platform introduced in late 2020 by 10X Genomics, the Chromium Single Cell Multiome ATAC and gene expression (10x Multiome), provides simultaneous gene expression and open chromatin measurements from the same cell at high throughput. This technology is well suited to identify gene regulatory networks by linking open chromatin regions with changes in gene expression, a task which is harder to perform when the two modalities are derived from separate groups of cells. However, very few datasets have been published to date using the 10x Multiome technology, and so how much information can be obtained by simultaneously profiling both modalities in the same cell remains an open question.

**Technology:** First, cells are purified and single nuclei are isolated, chromosomes are transpositioned. Next, ATAC and mRNA sequencing libraries are prepared with 10X Genomics Chromium microfluidic controller device where nuclei are partitioned and embedded in a droplet with a decorated gel bead with DNA 16nt 10X barcode that allows for pairing ATAC and mRNA signals to the same nuclei. mRNA is tagged with an 12nt Unique Molecular Identifier sequence (UMI), and a poly(dT)VN for poly-adenylated 3'ends. ATAC fragments are tagged with a Illumina primer sequence and an 8nt space sequence. All barcoded products are amplified in two rounds of PCR and then processed for sequencing. According to the

**Table 5. scNMT-seq dataset description, with of assay types, molecular modes, number of specimens, number of features and number of cells.**

| Dataset Identifier | Assay type | Modes | Data structure | # features | # cells |
|---|---|---|---|---|---|
| Mouse Gastrulation | RNA-seq | Transcripts | Matrix | 18345 | 2480 |
| | DNA Methylation | CpG islands | Matrix | 14080 | 986 |
| | | promoters | Matrix | 17179 | 986 |
| | | Gene bodies | Matrix | 17559 | 986 |
| | | DHS | Matrix | 6673 | 986 |
| | Chromatin accessibility | CpG islands | Matrix | 14824 | 1101 |
| | | promoters | Matrix | 18037 | 1103 |
| | | Gene bodies | Matrix | 17924 | 1105 |
| | | DHS | Matrix | 20082 | 1094 |

Chromium Single-Cell Multiome ATAC and gene expression assay product information, it has a flexible throughput of 500–10,000 nuclei per channel and up to 80,000 per run with a 65% recovery rate and low multiplet rate of <1% per 1000 cells (10Xgenomics.com).

**Landmark data**: 10X genomics has released a dataset of ~10k peripheral blood mononuclear cells (PBMCs) from a human healthy donor. Here we provide the RNA expression matrix and the binary matrix of ATAC fragments for each cell, quantified over a set of pre-computed peaks (Table 6). To access data in the SingleCellMultiModal package, call the scMultiome ("pbmc_10x") command. Relevant cell metadata is provided within the MultiAssayExperiment object. The overall dataset is 1.1 GB.

**RNA and spatial sequencing assays.    Purpose and goals**: The power of microscopy to resolve spatial information has been paired with single-cell sequencing to measure transcriptomic activity. These microscopy-based sequencing technologies capture a cell population's heterogeneous gene expression typically lost in bulk assays. Technologies like seqFISH(+) (sequential Fluorescence In Situ Hybridization), fluorescence in situ hybridization sequencing [7], Multiplexed error-robust fluorescence in situ hybridization (MERFISH) [39], Slide-seq [40,41] combine sequential barcoding with *in situ* molecular fluorescence probing, allowing the identification from tens to thousands of mRNAs transcripts while preserving spatial coordinates at micrometer resolution. We refer to this family of technologies as molecular-based spatial transcriptomics. Another family of spatial omics technologies can be described as spot-based; it includes the 10x Visium Spatial Gene Expression and Slide-seq [40]. In this family, the spatial coordinates are typically associated with barcoded spot-like identities, where the transcripts are amplified and sequenced. Currently, our package does not include any spot-based spatial transcriptomics dataset. The TENxVisiumData package [42] (available at https://github.com/HelenaLC/TENxVisiumData) contains several such datasets. See [43] for a comprehensive review of spatial transcriptomics technologies.

**Technology:** The seqFISH technology makes use of temporal barcodes to be read in multiple rounds of hybridization where mRNAs are labeled with fluorescent probes. During the hybridization rounds, the fluorescent probes are hybridized with the transcripts to be imaged with microscopy. Then they are stripped to be re-used and coupled with different fluorophores, during further rounds. In this case, the transcript abundance is given by the number of colocalizing spots per each transcript. The main differences between the technologies are due to the barcoding of RNAs. In seqFISH they are detected as a color sequence while in MERFISH the barcodes are identified as binary strings allowing error handling but requiring longer transcripts and more rounds of hybridizations [44].

**Landmark data:** The provided seqFISH dataset is designed on a mouse visual cortex tissue and can be retrieved in two different versions. Both versions include Single-cell RNA-seq and seqFISH data. Single-cell RNA-seq data in version 1.0.0 are part of the original paper [24] of 24057 genes in 1809 cells, while version 2.0.0 is a pre-processed adaptation of version 1.0.0 [22] where the authors analyzed it in order to provide the 113 genes in common with seqFISH data in 1723 cells. The provided seqFISH data are the same for both versions as part of their original paper [45,46] made of 1597 cells and 113 genes. The dataset is accessible via the `SingleCellMultiModal` Bioconductor package by using the `seqFISH(DataType="mouse_visual_cortex", version = "1.0.0")` function call, which returns a

**Table 6. 10X Multiome dataset descriptions, with assay types, molecular modes, number of features and number of cells.**

| Dataset Identifier | Assay type | Modes | Data structure | # features | # cells |
|---|---|---|---|---|---|
| Human PBMCs | RNA-seq | Gene expression | SingleCellExperiment | 36,549 | 10,032 |
| | Chromatin accessibility | Fragments over peaks | SingleCellExperiment | 108,344 | 10,032 |

**Table 7. seqFISH dataset descriptions, with assay types, molecular modes, specimens, dataset version provided, number of features and number of cells.**

| Dataset Identifier | Assay Type | Modes | Species | Data Structure | Version | # features | # cells |
|---|---|---|---|---|---|---|---|
| Mouse visual cortex | Single-cell RNA-seq | Transcripts | Mouse | SingleCellExperiment | 1.0.0 | 24057 | 1809 |
| | | | | | 2.0.0 | 113 | 1723 |
| | seqFISH | Spatial Transcriptomics | Mouse | SpatialExperiment | 1.0.0/2.0.0 | 113 | 1597 |

MultiAssayExperiment object with a SpatialExperiment object for the seqFISH data and a SingleCellExperiment object for the Single-cell RNA-seq data (Table 7).

**RNA and DNA sequencing assays. Purpose and goals**: Parallel genome and transcriptome sequencing (G&T-seq) of single-cells [8] opens new avenues for measuring transcriptional responses to genetic and genomic variation resulting from different allele frequencies, genetic mosaicism [47], single nucleotide variants (SNVs), DNA copy-number variants (CNVs), and structural variants (SVs). Although current experimental protocols are low-throughput with respect to the number of cells, simultaneous DNA and RNA sequencing of single-cells resolves the problem of how to associate cells across each modality from independently sampled single-cell measurements [48].

**Technology**: Following cell isolation and lysis, G&T-seq measures DNA and RNA levels of the same cell by physically separating polyadenylated RNA from genomic DNA using a biotinylated oligo-dT primer [49]. This is followed by separate whole-genome and whole-transcriptome amplification. Whole-genome amplification is carried out via multiple displacement amplification (MDA) or displacement pre-amplification and PCR (DA-PCR) for DNA sequencing, providing targeted sequencing reads or genome-wide copy number estimates. Parallel Smart-seq2 whole-transcriptome amplification is used for Illumina or PacBio cDNA sequencing, providing gene expression levels based on standard computational RNA-seq quantification pipelines. While pioneering technologies such as G&T-seq [8] and DR-seq [50] sequence both the DNA and RNA from single-cells, they currently measure only few cells (50–200 cells [51]) compared to assays that sequence DNA or RNA alone (1,000–10,000 cells [51]) such as Direct Library Preparation [52] or 10x Genomics Single-cell RNA-seq [53].

**Landmark data**: G&T-seq has been applied by Macaulay et al. [8] for parallel analysis of genomes and transcriptomes of (i) 130 individual cells from breast cancer line HCC38 and B lymphoblastoid line HCC38-BL, and (ii) 112 single cells from a mouse embryo at the eight-cell stage. Publicly available and included in the SingleCellMultiModal package is the mouse embryo dataset, assaying blastomeres of seven eight-cell cleavage-stage mouse embryos, five of which were treated with reversine at the four-cell stage of *in vitro* culture to induce chromosome mis-segregation. The dataset is stored as a MultiAssayExperiment [11] consisting of (i) a SingleCellExperiment [10] storing the single-cell RNA-seq read counts, and (ii) a RaggedExperiment [54] storing integer copy numbers as previously described [55] (Table 8). Although assaying only a relatively small number of cells, the dataset can serve as a prototype for benchmarking single-cell eQTL integration of DNA copy number and gene expression levels, given that Macaulay et al. [8] reported copy gains or losses with concomitant increases and decreases in gene expression levels.

**Table 8. G&T-seq dataset description, with assay types, molecular modes, number of specimens, number of features and number of cells.**

| Dataset Identifier | Assay Type | Mode | Species | Data Structure | Version | # features | # cells |
|---|---|---|---|---|---|---|---|
| E-ERAD-381 | RNA-seq | mRNA expression | Mouse | SingleCellExperiment | 1.0.0 | 23363 | 112 |
| | DNA-seq | Copy number | Mouse | RaggedExperiment | 1.0.0 | 2366 | 112 |

**Integrative analysis across modalities using data from SingleCellMultiModal.** Existing methods of integrative analysis of single-cell multimodal data have been recently reviewed [21]. Very briefly, some of the most popular current implementations are 1) the Seurat V4 R package which aims at vertical integration across several modal data types [56], 2) mixOmics [14] provides an extensive framework for data integration at molecular (P-integration, MINT [57]) and sample levels (N-integration, DIABLO [58]), and 3) Multi-Omics Factor Analysis, MOFA+ [15], a generalisation of Principal Components Analysis for inferring low-dimensional representation of multimodal data. Datasets provided by SingleCellMultiModal can be readily reshaped as input to any of these packages. We provide novel examples of such integrative analysis for exploratory visualization using SingleCellMultiModal datasets, produced within package documentation: MOFA+ [15] on the 10X Multiome dataset (Fig 4). For more information, see *Data Integration* Methods. In addition, we provide a sample analysis on the SCoPE2 dataset, which can be found in SingleCellMultimodal's package vignette.

## Methods

### SingleCellMultiModal data package

All datasets are distributed through the SingleCellMultiModal experimental data package in Bioconductor. This package employs ExperimentHub [59] for robust Cloud-based data download from AWS S3 buckets, with automatic local caching to avoid repetitive downloads. These methods are described in detail elsewhere for application to The Cancer Genome Atlas and cBioPortal [60]. Briefly, metadata and individual omics datasets are stored in ExperimentHub as simple core Bioconductor objects such as matrix, SparseMatrix, SingleCellExperiment, and RaggedExperiment. A simple user-facing convenience function is provided for each dataset that retrieves all necessary individual components, assembles a MultiAssayExperiment object [11], and returns this to the user. For very large matrices we employ HDF5 and MTX on-disk representation. Methods for users to access these datasets are documented in the SingleCellMultiModal package vignette and functions manual.

All datasets provide a cell-type annotation either provided by the authors of the original publications or based on external knowledge, except for ECCITEseq and G&T-seq for which none was available (Table 9). In particular, the SCoPE2, seqFISH, and scNMT datasets include the cell type labels provided by the authors of the original publication, while scMultiome includes a cell type label inferred by manually inspecting cluster markers. While the CITEseq dataset does not formally have cell type labels, the ADT data for cell surface markers typically used in immunology allow for manual gating of cell types (see e.g., S3 Fig of [3]). We used this additional information to define cell type labels that are independent of scRNA-seq data. We note that this dataset has been previously used for method benchmarking by looking for maximal correspondence between scRNA-seq derived cell type labels and ADT surface markers, which can be considered as gold standard (see e.g. [61]). A dedicated function (getCellGroups) has been added to the SingleCellMultiModal package to help the user to add their own cell annotations based on manual gating. An overview of the cell type annotation for all the dataset can be easily retrieved through the ontomap function, which also standardizes cell names from the Cell Ontology [62] (first preference) or NCI Thesaurus [63] to facilitate cross-dataset comparisons.

Furthermore, each dataset includes information on the quality of the cells: for some datasets, the package provides a pre-filtered object, in which low-quality cells were already excluded by the original authors; for others, we provide an option to retrieve either the filtered dataset, which include only cells that pass a quality control step, or the full dataset along with an indication, for each cell, of whether it passes the QC filters, defined following state-of-the-art best practices (Table 9).

**From 10X Genomics**

**A** Raw seq data: FASTQ and sparseMatrix available
- RNA-seq
- ATAC-seq

Pre-processing from *SCMM*

1. Data import in alternative formats:
   - dgCMatrix (MTX)
   - HDF5 on-disk
2. Integration of RNA-seq, ATAC-seq, metadata using `MultiAssayExperiment`
3. Automatic caching via ExperimentHub

**In *SCMM* package**

**B** Call:
`scMultiome("pbmc_10x")` provides one `MAE` object with two experiments

```
A MultiAssayExperiment object of 2 listed
 experiments with user-defined names and respective classes.
 Containing an ExperimentList class object of length 2:
 [1] atac: SingleCellExperiment with 108344 rows and 10032 columns
 [2] rna: SingleCellExperiment with 36549 rows and 10032 columns
```

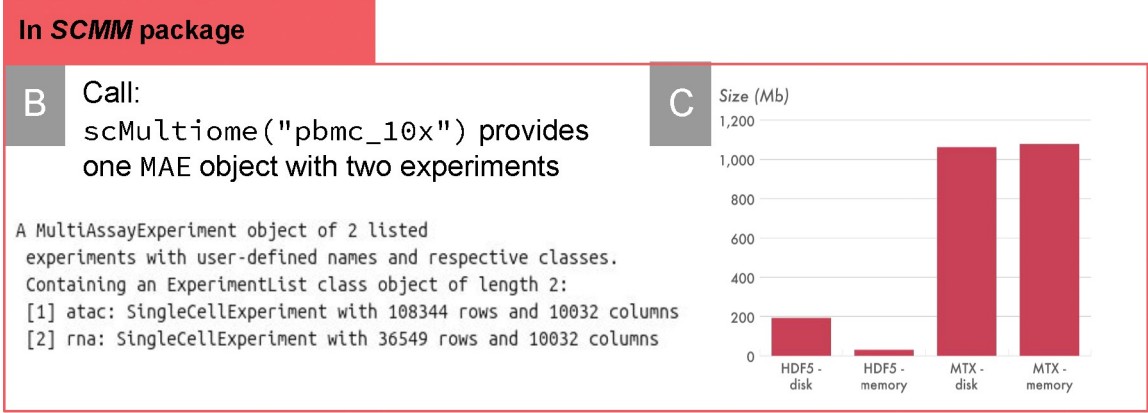

**MOFA+ analysis in BioC**

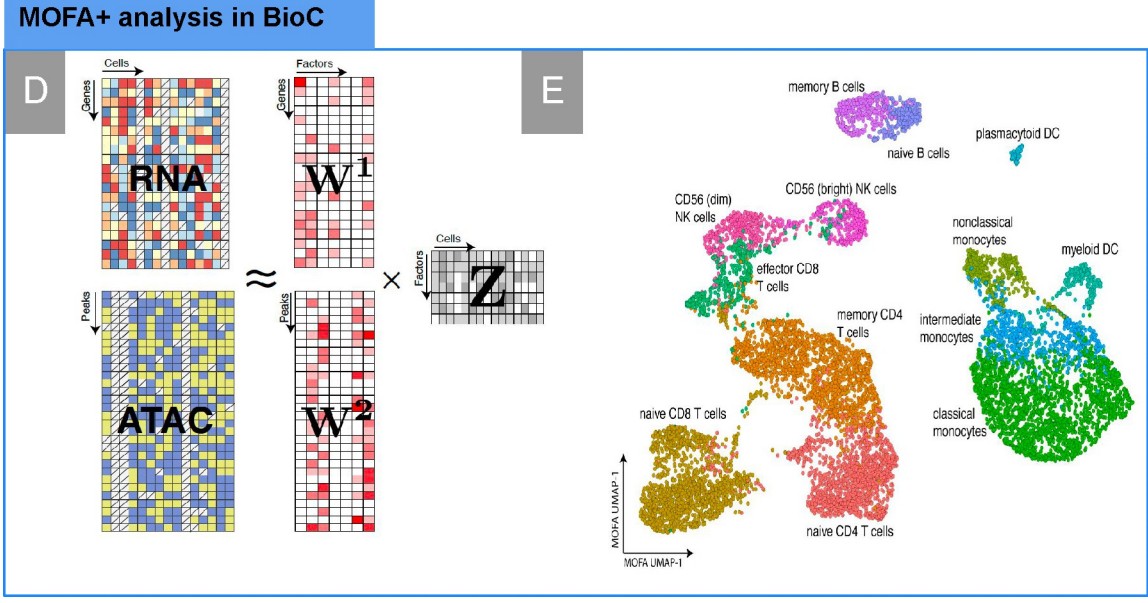

**Fig 4. Summary of example integration using the 10X Genomic Multiome data from the S*ingleCellMultiModal* package. A**): *(left)* from the 10X Genomic Multiome data resource, a sparse matrix and FASTA datasets provided, *(right)* pre-processing steps of source data required to work with of Multiome dataset provided in S*ingleCellMultiModal* **B):** MultiAssayExperiment (MAE) object returned when called **C):** Bar graph of the alternative data import sizes **D), E)** data integration of 10x Genomic Multiome dataset, combining the chromatin accessibility data with the transcriptome data using MOFA+. **D):** RNA-seq and ATAC-seq matrices used for weight factored analysis, **E):** UMAP cluster of cell types based on factor analysis. For more detail on the analysis, see the Methods and the *SingleCellMultiModal* package vignette. Other datasets can be represented similarly: raw data processing and integration of data modes occur upstream of the SingleCellMultiModal package, users invoke a single command that creates a MultiAssayExperiment integrating appropriate memory-efficient objects, which are applied directly to downstream R/Bioconductor analyses.

**Table 9. Dataset information on cell filtering and annotations.** Specifies if the quality controls on the cells of each dataset have been already performed or if it is present in the retrieved object. Column Version QC indicates the version of the dataset where QC is available. Cell Annotation column indicates which datasets have a ground-truth available.

| EXPERIMENTAL ASSAY | DATATYPE NAME | CELL QC | VERSION QC | CELL ANNOTATION | CELL ANNOTATION COLUMN |
|---|---|---|---|---|---|
| G&T-seq | mouse_embryo_8_cell | Filtered as in [8] | 1.0.0 | absent | NA |
| CITE-Seq | cord_blood | Based on [64,65] | 1.0.0 | Manual gating on ADT markers | celltype |
| ECCITE-Seq | peripheral_blood | Based on [64,65] | 1.0.0 | absent | NA |
| scNMT-seq | mouse_gastrulation | Based on [5] | 2.0.0 | Provided by (22) | lineage |
| 10X Multiome | pbmc_10x | Based on (66) | 1.0.0 | Manual curation based on markers | celltype |
| seqFISH | mouse_visual_cortex | unavailable | 1.0.0 | Provided by [21] | class |
| | | Filtered as in [21] | 2.0.0 | | |
| SCoPE2 | macrophage_differentiation | Filtered as in [9] | 1.0.0 | Provided by [9] | celltype |

## CITE-Seq and ECCITE-Seq dataset

The CITE-Seq contains two modalities of cord blood mononuclear cells, the transcripts (scRNA) and the cell surface proteins (scADT) measured and preprocessed as described in the CITE-Seq landmark paper [3]. The PBMC UMI counts together with the Centered Log Ratio (CLR) transformed counts for the scADT were retrieved from the GEO repository with accession number GSE100866 and then loaded in R to be transformed in matrix format and then be loaded as separate assays of a MultiAssayExperiment object. This latter object can be retrieved from the SingleCellMultiModal package with the function call: CITEseq(DataType = "cord_blood", dry.run = FALSE).

Because this dataset provides a mixture of human and mouse cells, we applied k-means clustering combined with hierarchical clustering on the log transformed scRNA counts to identify then classify human cells, approximately reconstructing the population of cells that the landmark paper reported through manual gating on the ADT assay. Despite a few unidentified cells, most of the subgroups of cells are identified, i.e., Natural Killers, Precursors, CD4 T-cells, CD8 T-cells, B-cells, Monocytes CD14+ and Monocytes CD16+. The code for cell filtering and cell manual gating has been released inside the script folder of the SingleCellMultiModal package. Additionally, two functions (getCellGroups and addCTLabels) for the manual gating are now exported by the package to help users identify their own cell populations.

The ECCITE-Seq has three modalities of the of peripheral blood mononuclear cells, (scRNA), the cell surface proteins (scADT) and the Hashtagged Oligo (scHTO) measured and preprocessed as described in the ECCITE-Seq landmark paper [4].

The PBMC modalities for the cutaneous T-cell lymphoma (CTCL) and controls (CTRL) were retrieved in TXT format from the GEO repository with accession number GSE126310 and then loaded in R to be transformed in matrix and data.frame format and then be loaded as separate assays of a MultiAssayExperiment object. The CRISPR perturbed scRNAs data are stored as data.frame in the object metadata to keep their original long format. This latter object can be retrieved from the SingleCellMultiModal with the function call:
CITEseq(DataType = "peripheral_blood", dry.run = FALSE).

## Visual cortex seqFISH dataset

The seqFISH dataset has two different modalities, the spatial transcriptomics (seqFISH) and the single-cell RNA-seq, in two different versions. The main difference between the two versions are in the Single-cell RNA-seq counts data which in version 1.0.0 are provided as

downloaded in CSV format from the GEO repository with accession number GSE71585, while the version 2.0.0 is a processed dataset [46] where only the genes with correspondence in the seqFISH dataset have been preserved. Methods of pre-processing are described at https://github.com/BIRSBiointegration/Hackathon. In both versions the seqFISH dataset is the processed version [46] as downloaded from https://cloudstor.aarnet.edu.au/plus/s/ZuBIXuzuvc9JMj3. Processed version of the seqFISH data were downloaded as TXT format for the coordinates (fcortex.coordinates.txt), as TSV format for the cell annotated labels (seqfish_labels.tsv) and TXT format for the counts (seqfish_cortex_b2_testing.txt). The data constitute a SpatialExperiment object with the counts as assay, the cell labels as colData and the coordinates stored as spatialData.In the same way, the processed Single-cell RNA-seq data were downloaded as TXT format for the counts (tasic_training_b2.txt), as TSV format for the cell annotated labels (tasic_labels.tsv) to build a SingleCellExperiment object with the counts as assay and the cell labels as colData.

Finally, the SingleCellExperiment and the SpatialExperiment have been loaded into a MultiAssayExperiment object as two different assays. The MultiAssayExperiment object can be retrieved with the function call, for example:

seqFISH(DataType = "mouse_visual_cortex",dry.run = FALSE, version = "2.0.0")

## Mouse gastrulation scNMT dataset

Preprocessing methods are described in full by Argelaguet *et al.* [23]. Briefly, RNA-seq libraries were aligned to the GRCm38 mouse genome build using HiSat235 (v.2.1.0). Gene expression counts were quantified from the mapped reads using featureCounts [66] with the Ensembl 87 gene annotation [67]. The read counts were log-transformed and size-factor adjusted using scran normalization [68]. Bisulfite-seq libraries were aligned to the bisulfite converted GRCm38 mouse genome using Bismark [69]. Endogenous CpG methylation was quantified over ACG and TCG trinucleotides and GpC chromatin accessibility over GCA, GCC and GCT trinucleotides. Note that for GCG trinucleotides it is not possible to distinguish endogenous CpG methylation from induced GpC methylation. In addition, CGC positions were discarded because of off-target effects of the GpC methyltransferase enzyme [70].

For each CpG site in each cell we obtained binary methylation calls and for each GpC site in each cell we obtained binary accessibility calls. Notice that binary readouts is an exclusive property of single-cell bisulfite sequencing data, as for the vast majority of sites only one allele is observed per cell. This contrasts with bulk bisulfite sequencing data, where each dinucleotide typically contains multiple reads originating from different cells.

Finally, we quantified DNA methylation and chromatin accessibility over genomic features by assuming a binomial model is assumed for each cell and feature, where the number of successes is the number of methylated CpGs (or GpCs) and the number of trials is the total number of CpGs (or GpCs) that are observed within the specific cell and genomic feature. Here, We quantified DNA methylation and chromatin accessibility rates over CpG islands, gene promoters, gene bodies and DNAse hypersensitive sites. All these data modalities were compiled together with the RNA expression into a MultiAssayExperiment object. The dataset can be loaded from within the SingleCellMultiModal `package` by the function call scNMT("mouse_gastrulation", dry.run = FALSE, version = "2.0.0"). Code with the data processing pipeline is available in https://github.com/rargelaguet/scnmt_gastrulation.

## 10X multiome dataset

PBMCs were extracted from a healthy donor after removing granulocytes through cell sorting. The dataset was downloaded as a CellRanger ARC output from https://support.10xgenomics.

com/single-cell-multiome-atac-gex/datasets/1.0.0/pbmc_granulocyte_sorted_10k, which includes the gene expression matrix and the chromatin accessibility matrix quantified over ATAC peaks. The dataset included 11,909 cells with a median of 13,486 high-quality ATAC fragments per cell and a median of 1,826 genes expressed per cell. Data processing details, including the peak calling algorithm, can be found in https://support.10xgenomics.com/ single-cell-multiome-atac-gex/software/pipelines/latest/what-is-cell-ranger-arc. The dataset is provided as a MultiAssayExperiment [11] consisting of two SingleCellExperiment [10], one containing the single-cell RNA-seq read counts, and the other containing the binary ATAC peak matrix. The dataset can be loaded from within the SingleCellMultiModal `package` by the function call scMultiome("pbmc_10x", dry.run = FALSE).

## Macrophage differentiation SCoPE2 dataset

The macrophage differentiation project contains two datasets: single-cell RNA-seq data and MS-SCP data. Upstream processing is described in detail in the SCoPE2 landmark paper [9]. Briefly, for the Single-cell RNA-seq dataset, the authors used CellRanger to align the reads and to build the UMI count matrices. Based on cell QC and manual inspection, they discarded cells containing less than $10^4$ UMI barcodes. The resulting tables for two technical replicates were deposited in a GEO repository with accession GSE142392. For the MS-SCP dataset, the authors followed the workflow described in Fig 3, with identification and quantification steps performed using the MaxQuant software and additional protein quantification using a custom R script available on GitHub (https://github.com/SlavovLab/SCoPE2).

 We retrieved the single-cell RNA-seq dataset from the GSE142392 repository. The MS-SCP data and annotations were retrieved from CSV files available at the authors' website (https:// scope2.slavovlab.net/docs/data). We formatted the Single-cell RNA-seq and the MS-SCP data as two separate `SingleCellExperiment` objects without further processing. Because the Single-cell RNA-seq data is relatively large, it is stored as a sparse matrix using the HDF5 data format. We combined the two data objects in a single MultiAssayExperiment object. This latter object can be queried from the SingleCellMultimodal package with the function call SCoPE2 ("macrophage_differentiation", dry.run = FALSE).

## G&T-seq dataset

Raw sequencing data was obtained from the European Nucleotide Archive (ENA [71], accession PRJEB9051). The data was downloaded in fastq files for whole-genome and whole-transcriptome paired-end sequencing data for 112 mouse embryo cells. The data was processed as described in the step-by-step protocol of Macaulay et al. [49]. Preprocessing and mapping of genome sequencing data was carried out following steps 78–84 of the protocol of Macaulay et al. [49], using Rsubread [72] for read trimming, alignment to the mm10 mouse reference genome, and removal of PCR-duplicate reads. DNA copy-number profiling was carried out following steps 85–87, using bedtools [73] to convert BAM to BED files, and subsequently applying Ginkgo [74] for copy number determination. Preprocessing and mapping of transcriptome sequencing data was carried out following steps 94–96, using Rsubread [72] for read trimming and alignment to the mm10 mouse reference genome. Read counts for each gene were obtained using the featureCounts [66] function of the Rsubread package. The dataset is provided as a MultiAssayExperiment [11] consisting of (i) a SingleCellExperiment [10] storing the single-cell RNA-seq read counts, and (ii) a RaggedExperiment [54] storing integer copy numbers as previously described [55]. The dataset can be loaded from within the SingleCell-MultiModal package by the function call GTseq(dry.run = FALSE).

### Data integration of the 10x multiome data set

For the integration of the 10x Multiome dataset we used MOFA+ [15] to obtain a latent embedding with contribution from both data modalities. The RNA expression was normalised using scran [68], followed by feature selection of the top 2000 most variable genes. The chromatin accessibility was normalised using TFIDF, followed by feature selection of the top 10,000 peaks with the highest mean accessibility. The MOFA model was trained with $K = 15$ factors using default options. To obtain a non-linear embedding we applied UMAP [75] on the MOFA factors.

## Discussion

Experimental data packages providing landmark datasets have historically played an important role in the development of new statistical methods in Bioconductor, from the classic acute lymphocytic leukemia (*ALL*) microarray dataset [76] to the *HSMMSingleCell* single-cell RNA-seq dataset [77], as well as packages providing more extensive curated selections of standardized datasets in a specific realm [78]. Such packages greatly lower the barrier of access to relevant data for developers of scientific software, and provide a common testing ground for development and benchmarking. We present the *SingleCellMultiModal* Bioconductor experimental data package, to distribute landmark single-cell multimodal datasets in pre-integrated immediately usable forms, utilizing standard Bioconductor data structures. Multimodal datasets are serialized as a *MultiAssayExperiment* object by a single command, without requiring users to perform data wrangling to link multiple 'omics profiles or to manage cells with incomplete data. We provide curated landmark datasets for a selection of key single-cell multimodal assays that will serve as benchmarks for the development and assessment of appropriate analysis methods in R/Bioconductor. We provide a brief review of the assays provided for the purpose of providing essential background to developers of statistical and bioinformatic methods, a summary of the data contained in each dataset, and examples of minimal code needed to access each dataset in an R/Bioconductor session. Methods of statistical analysis are reviewed in a recent complimentary paper [22].

Single-cell RNA-seq analysis methods in Bioconductor are well developed and widely used [10], setting the stage for new development in single-cell multimodal data analysis that will be facilitated by the SingleCellMultiModal experimental data package. Areas of active research include integrative systems biology across data modes, spatial statistics on high-dimensional data, dimension reduction and clustering [14], cell identification, multimodal batch correction, and new data structures for representation and analysis of large and spatially resolved single-cell multimodal data. These areas of research and their software products will be facilitated and made more interoperable by the easily accessible and uniformly represented data provided by this work.

## Author Contributions

**Conceptualization:** Kelly B. Eckenrode, Dario Righelli, Marcel Ramos, Ricard Argelaguet, Christophe Vanderaa, Ludwig Geistlinger, Aedin C. Culhane, Laurent Gatto, Martin Morgan, Davide Risso, Levi Waldron.

**Data curation:** Kelly B. Eckenrode, Dario Righelli, Marcel Ramos, Ricard Argelaguet, Christophe Vanderaa, Ludwig Geistlinger.

**Formal analysis:** Dario Righelli, Marcel Ramos, Ricard Argelaguet, Christophe Vanderaa, Ludwig Geistlinger.

**Investigation:** Kelly B. Eckenrode, Marcel Ramos.

**Methodology:** Dario Righelli, Marcel Ramos, Ricard Argelaguet, Christophe Vanderaa, Ludwig Geistlinger.

**Project administration:** Kelly B. Eckenrode, Dario Righelli.

**Resources:** Dario Righelli, Marcel Ramos, Ricard Argelaguet, Christophe Vanderaa, Ludwig Geistlinger.

**Software:** Dario Righelli, Marcel Ramos, Ricard Argelaguet, Christophe Vanderaa, Ludwig Geistlinger.

**Supervision:** Aedin C. Culhane, Laurent Gatto, Vincent Carey, Martin Morgan, Davide Risso, Levi Waldron.

**Validation:** Dario Righelli, Marcel Ramos, Ricard Argelaguet, Ludwig Geistlinger.

**Visualization:** Kelly B. Eckenrode, Dario Righelli, Ricard Argelaguet.

**Writing – original draft:** Kelly B. Eckenrode, Dario Righelli, Marcel Ramos, Ricard Argelaguet, Christophe Vanderaa, Ludwig Geistlinger, Aedin C. Culhane.

**Writing – review & editing:** Kelly B. Eckenrode, Dario Righelli, Ricard Argelaguet, Christophe Vanderaa, Ludwig Geistlinger, Aedin C. Culhane, Laurent Gatto, Vincent Carey, Davide Risso, Levi Waldron.

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
