## [Decision Letter · Decision Letter 0]

13 Mar 2023

Dear Associate Professor Waldron,

Thank you very much for submitting your manuscript "Curated Single Cell Multimodal Landmark Datasets for R/Bioconductor" for consideration at PLOS Computational Biology.

As with all papers reviewed by the journal, your manuscript was reviewed by members of the editorial board and by several independent reviewers. In light of the reviews (below this email), we would like to invite the resubmission of a significantly-revised version that takes into account the reviewers' comments.

We cannot make any decision about publication until we have seen the revised manuscript and your response to the reviewers' comments. Your revised manuscript is also likely to be sent to reviewers for further evaluation.

Sincerely,

Mingyao Li

Academic Editor

PLOS Computational Biology

William Noble

Section Editor

PLOS Computational Biology

Reviewer's Responses to Questions

**Comments to the Authors:**

Reviewer #1: The authors collected multimodal single-cell datasets from multiple human tissues and build a resource named SingleCellMultiModal. The landmark datasets and the Bioconductor package presented here will be useful for new computational method development. However, I have two concerns about the manuscript.

1. Collecting the right datasets is important for computational method development. One challenging point is that most datasets do not have cell type label information. Thus, it is difficult to evaluate the newly developed methods. There are some datasets that have great quality and have been analyzed in literatures by some great methods or have been manually annotated. Collection of those datasets will be extremely useful for method development. The impact of the software/manuscript will be significantly increased if datasets with annotated cell labels will be included.

2. Because of technical reason, there are always have some bad quality cells. Including too much bad quality cells will affect the performance of the computational methods. It would be helpful for users if a reasonable quality score is included.

Reviewer #2: The manuscript ‘Curated Single Cell Multimodal Landmark Datasets for R/Bioconductor’ described an effort to compile landmark datasets from various of single-cell multi-modal technologies in one place. Specifically, the authors created an R/Bioconductor package, SingleCellMultiModal that allows the easy access to these datasets. This is a valuable resource for method developer and for experimentalist who might be thinking about generating the multi-modal datasets.

Major concern:

1. For method developers who want to test the method using a publicly available dataset, they will need to have the ground-truth annotation associated with the data. For most of the datasets described in the manuscript, a common ground-truth information one would need is the cell type annotation. It’d be clearer if the authors have more explicit description on what metadata is provided for each dataset.

2. For some of the data types, such as 10X multiome and seqFISH datasets, there are additional datasets critical for the analysis. For example, the fragment file (listing the open fragment region sequenced per cell) for 10X multiome, and the fluorescence images for seqFISH. How would these data be distributed to the users?

3. Currently, it is difficult to conceptualize how an object from the SingleCellMultiModal package looks like. I think adding a more detailed description and potentially a graphical illustration of the object under the ‘Summary of landmark datasets in SingleCellMultiModal’ heading will be beneficial for readers.

Minor concern:

1. For Figure 1A and 1B, enlarge the font. Currently, I need to zoom in quite a lot to see the text in the figures. For the x-axis labels, could enlarge the font and rotate them to avoid stacking.

2. There is a typo in the Figure 4 legend. Should be ‘(right) pre-processing steps of source data’.

**Have the authors made all data and (if applicable) computational code underlying the findings in their manuscript fully available?**

Reviewer #1: Yes

Reviewer #2: Yes

PLOS authors have the option to publish the peer review history of their article (what does this mean?). If published, this will include your full peer review and any attached files.

Reviewer #1: **Yes: **Zhana Duren

Reviewer #2: No
---

## [Decision Letter · Decision Letter 1]

3 Jul 2023

Dear Associate Professor Waldron,

We are pleased to inform you that your manuscript 'Curated Single Cell Multimodal Landmark Datasets for R/Bioconductor' has been provisionally accepted for publication in PLOS Computational Biology.

Best regards,

Mingyao Li

Academic Editor

PLOS Computational Biology

William Noble

Section Editor

PLOS Computational Biology

Reviewer's Responses to Questions

**Comments to the Authors:**

Reviewer #1: Authors have addressed all my comments and I have no further comments.

Reviewer #2: Thanks for answering all the comments. I do not have further questions. Great work!

**Have the authors made all data and (if applicable) computational code underlying the findings in their manuscript fully available?**

Reviewer #1: Yes

Reviewer #2: Yes

PLOS authors have the option to publish the peer review history of their article (what does this mean?). If published, this will include your full peer review and any attached files.

Reviewer #1: **Yes: **Zhana Duren

Reviewer #2: **Yes: **Michelle Y. Y. Lee

---

## [Editor Report · Acceptance letter]

8 Aug 2023

PCOMPBIOL-D-22-01713R1 

Curated Single Cell Multimodal Landmark Datasets for R/Bioconductor

Dear Dr Waldron,

I am pleased to inform you that your manuscript has been formally accepted for publication in PLOS Computational Biology. Your manuscript is now with our production department and you will be notified of the publication date in due course.

With kind regards,

Timea Kemeri-Szekernyes
